# Psychological Treatments for Depression in Adolescents: More Than Three Decades Later

**DOI:** 10.3390/ijerph18094600

**Published:** 2021-04-26

**Authors:** Javier Méndez, Óscar Sánchez-Hernández, Judy Garber, José P. Espada, Mireia Orgilés

**Affiliations:** 1Department of Personality, Assessment and Psychological Treatment, University of Murcia, 30100 Murcia, Spain; 2Department of Developmental and Educational Psychology, University of Murcia, 30100 Murcia, Spain; oscarsh@um.es; 3Department of Psychology and Human Development, Vanderbilt University, Nashville, TN 37302, USA; judy.garber@vanderbilt.edu; 4Department of Health Psychology, Miguel Hernández University, 03202 Elche, Spain; jpespada@umh.es (J.P.E.); morgiles@umh.es (M.O.)

**Keywords:** adolescents, depression, psychological treatments, qualitative review

## Abstract

Depression is a common and impairing disorder which is a serious public health problem. For some individuals, depression has a chronic course and is recurrent, particularly when its onset is during adolescence. The purpose of the current paper was to review the clinical trials conducted between 1980 and 2020 in adolescents with a primary diagnosis of a depressive disorder, excluding indicated prevention trials for depressive symptomatology. Cognitive behavioral therapy (CBT) is the pre-eminent treatment and is well established from an evidence-based treatment perspective. The body of research on the remaining treatments is smaller and the status of these treatments is varied: interpersonal therapy (IPT) is well established; family therapy (FT) is possibly effective; and short-term psychoanalytic therapy (PT) is experimental treatment. Implementation of the two treatments that work well—CBT and IPT—has more support when provided individually as compared to in groups. Research on depression treatments has been expanding through using transdiagnostic and modular protocols, implementation through information and communication technologies, and indicated prevention programs. Despite significant progress, however, questions remain regarding the rate of non-response to treatment, the fading of specific treatment effects over time, and the contribution of parental involvement in therapy.

## 1. Depression in Adolescence: A Public Health Problem

Depression is a major public health concern; it is the most disabling single disorder, contributing to 7.2% of the overall burden of disease in Europe [1]. Depression is common, recurrent, sometimes chronic, and often comorbid with anxiety and substance and alcohol use problems.

Epidemiological studies of the prevalence of depression in adolescents differ depending on the diagnostic criteria used (e.g., Diagnostic and Statistical Manual of Mental Disorders, DSM; International Statistical Classification of Diseases and Related Health Problems, ICD; Research Diagnostic Criteria, RDC), the period covered (e.g., last month, last year, lifetime), the nature of the sample (e.g., school, pediatric, clinical), the age range (e.g., early, middle, late adolescence), the method of assessment (e.g., interview, questionnaires, rating scales), the informant (e.g., adolescent, parent, clinician), and other aspects such as country or research design. In the United States, the National Comorbidity Survey-Adolescent Supplement (NCS-A) found a lifetime and annual prevalence of 11.0% and 7.5%, respectively, with a representative sample of 10,123 adolescents, aged 13 to 18 years old [2]. The Saving and Empowering Young Lives in Europe (SEYLE) study, conducted in ten European countries plus Israel, with a sample of 12,395 adolescents, mean age 14.81 years, found that prevalence ranged from 7.1% to 19.4%. The percentage per country in increasing order was Hungary 7.1%, Austria 7.6%, Romania 7.6%, Estonia 7.9%, Ireland 8.5%, Spain 8.6%, Italy 9.2%, Slovenia 11.4%, Germany 12.9%, France 15.4%, and Israel 19.4% [3]. The Young Minds Matter, the second Australian Child and Adolescent Survey of Mental Health and Wellbeing study, with 6310 children and adolescents, found a prevalence of depression during the previous twelve months in youths 12 to 17 years old, of 5.0%; 4.7% if the informant was the parent or caregiver and 7.7% if the adolescent (aged 11–17 years) was the informant [4].

Depression is one of the main risk factors for suicide in adolescence. Among adolescents aged 15–19 years old, suicide is the third leading cause of death; in 2016, an estimated 62,000 adolescents died from self-harm [5]. In 2018, suicide was one of the three leading causes of death in Spain in the 15 to 19 age group: 18.2% traffic accidents, 17.7% tumors, and 17.0% suicide [6].

Comorbidity is common in adolescents with depression. In 1990, Fleming and Offord concluded “that ‘pure’ depression in children and adolescents is a rare entity” [7], p. 575. Clinical and epidemiological studies show that 40–80% of children and adolescents with depression have at least one other disorder and 20–50% have two or more comorbid disorders. Adolescent depression is associated with impairment in academic performance and family and peer relationships, and with a wide range of other disorders and problems, including anxiety, alcohol and drug abuse, risky sexual behavior, hyperactivity, oppositional behavior, antisocial behavior, delinquency, eating disorders, and self-injury [8].

Relapse and recurrence are common in adolescent depression. The Treatment for Adolescents with Depression Study (TADS) Team found that 88 of 189 adolescents (46.6%), who had recovered from depression, had a recurrence within 63 months, once (39.2%), twice (6.3%) and even three (1.1%) times. The mean time from recovery to first recurrence was 22.3 months, range 2–55 months, with the following cumulative rates every six months: 12.5% at half a year, 26.1% at one year, 40.9% at one and a half years, 61.3% at two years, 77.3% at two and a half years, and 84.9% at three years [9]. The recurrence rate in adulthood of depression that first appeared during adolescence is estimated at 60–70% [10].

Research on the efficacy of psychological treatments gained momentum in 1980, when the American Psychiatric Association settled the controversy about the nature of childhood and adolescent depression by clarifying that “the essential features of a major depressive episode are similar in infants, children, adolescents and adults” [11], p. 211. That year, the first trial on the treatment of childhood depression was published [12], and six years later, the first study of the treatment of depression in adolescents was published, involving 30 school children, mean age 15.65 years [13]. The authors argued that if adolescent depression is similar to adult depression, then treatment applied to the adult population, adapted to the level of adolescent development, would be effective in overcoming depression in adolescents. They considered cognitive behavioral therapy (CBT) the treatment of choice, which they compared with progressive relaxation training, and a wait-list (WL) control. Although relaxation is not a specific therapy for depression, they justified its inclusion as an active treatment because of the link between stress and depression. Contrary to their hypothesis, they found no differences between CBT and relaxation, with 83% of the CBT group and 75% of the relaxation group free of depression at the end of treatment, whereas all adolescents in the WL group continued to meet the criteria for depression.

The first randomized controlled trial (RCT) in adolescents diagnosed with depression was conducted in 1990 [14]; since then, the number of clinical trials has multiplied, which has substantially improved our knowledge about the efficacy of psychological treatments of depression in adolescents. Nevertheless, although there are treatments of proven efficacy, “research-based treatments are rarely used in clinical practice” [15], p. 765, highlighting a gap between clinical science and clinical practice.

Depression is often underdiagnosed during adolescence. Moreover, although about 20% of adolescents experience an episode of major depression, only a few receive evidence-based treatments [16]. The objectives of the current review are, first, to analyze, from a historical perspective, the current state of psychological treatments for adolescent depression according to the criteria used to evaluate evidence-based psychological treatments for children and adolescents (Table 1) [17]. The second aim is to provide health professionals with guidelines for choosing the most appropriate treatment based on the current evidence. Thus, this review addresses the question regarding the current evidence of the efficacy of various psychological treatments for depression in adolescents.

## 2. Psychological Treatments for Depression in Adolescents

The criteria for including a trial in the present review were: (a) that the mean age of participants was between 12 and 18 years old; (b) a primary diagnosis of depression; (c) randomized controlled trial; (d) valid and reliable depression assessment measures; (e) comparison of at least one psychological treatment with another psychological, pharmacological, and/or placebo treatment, and/or waitlist (WL); and (f) publication in a peer-reviewed journal; (g) between 1980 and September 2020.

Exclusion criteria were: (a) mean age of participants was less than 12 or more than 18 years; (b) depressive symptomatology only, without a diagnosis of depression; (c) open trial or case study; (d) trials testing medication alone; (e) preventive interventions; and (f) publication in media other than peer-reviewed journals (handbooks, conference proceedings, etc.); (g) prior to 1980 or after September 2020.

We used several strategies to identify trials: (a) searches in the PsycINFO, PubMed, ERIC, Web of Science and CSIC-ISOC databases; (b) websites of institutions: Division 53. Society of Clinical Child and Adolescent Psychology, of the American Psychological Association (APA) (Retrieved 13 April 2021 from https://effectivechildtherapy.org/concerns-symptoms-disorders/); National Institute for Health and Care Excellence (NICE) (Retrieved 13 April 2021 from https://www.nice.org.uk/guidance/ng134); National Health System (Sistema Nacional de Salud, SNS) of Spain (Retrieved 13 April 2021 from https://portal.guiasalud.es/gpc/depresion-infancia/); and (c) the retrieval of primary studies from systematic reviews and meta-analyses.

We identified 123 potential trials, published between 1986 and 2020, of which 96 were excluded [13,16,18,19,20,21,22,23,24,25,26,27,28,29,30,31,32,33,34,35,36,37,38,39,40,41,42,43,44,45,46,47,48,49,50,51,52,53,54,55,56,57,58,59,60,61,62,63,64,65,66,67,68,69,70,71,72,73,74,75,76,77,78,79,80,81,82,83,84,85,86,87,88,89,90,91,92,93,94,95,96,97,98,99,100,101,102,103,104,105,106,107,108,109,110,111] (see Table 2). The main reason for exclusion was that they were prevention rather than treatment trials, mostly with indicated samples (adolescents with depressive symptoms, but without a diagnosis of depression). Other reasons for exclusion were the mean age of participants below 12 years (preadolescents) or above 18 years (youths), noncompliance with methodological standards (e.g., nonrandom assignment, open trial), and recruitment of heterogeneous samples with depression or other disorders. The 27 selected adolescent depression treatment trials [14,112,113,114,115,116,117,118,119,120,121,122,123,124,125,126,127,128,129,130,131,132,133,134,135,136,137] resulted in 46 studies in which a psychological treatment was compared with another psychological or pharmacological treatment, or with an active control or WL condition. CBT was the most investigated treatment with 22 trials (81%). Research on other treatments is rather scarce, with four trials of family therapy (FT), three of interpersonal therapy (IPT), and a single trial of psychoanalytic therapy (PT).

The 27 RCTs reviewed involved 3501 adolescents, mean age 15 years, 66.6% girls, from families with varied socioeconomic status and structure, and of diverse ethnicity (see Table 3). The adolescents had major depressive disorder (87.4%) and/or persistent depressive/dysthymic disorder (10%), unspecified depressive disorder (0.9%), or another depressive disorder (5%). In ten trials, with 1391 participants, suicidality was reported: 36.8% of adolescents had current suicidal ideation and/or 35% had a history of suicide attempts. In sixteen trials with 2483 adolescents, a wide range of comorbid disorders were reported: 43.1% anxiety disorders, 28.3% disruptive and conduct disorders, 17% attention deficit hyperactivity disorder, and 26.6% had other disorders. The trial by Szigethy and colleagues [130] departs from the above, because the comorbidity of the participating children and adolescents was inflammatory bowel disease, Crohn’s disease (75%) or ulcerative colitis (25%).

### 2.1. Cognitive–Behavioral Therapy

Considering the large number of studies of CBT and the fact that definitions of evidence-based treatment emphasize the accumulation of positive findings, we focused on the analysis of trials with positive effects. Table 4 presents the RCTs that evaluated treatments for depression in adolescents included in the current review.

The first trials compared CBT with another psychological treatment, with active controls, or WL. Approximately fifteen years ago, research began to investigate the effectiveness of CBT combined with antidepressants; additionally, WL was used less and was replaced by treatment as usual (TAU).

#### 2.1.1. Cognitive–Behavioral Therapy Alone

Lewinsohn and colleagues [14] conducted a pioneering trial with two objectives: (1) to test the efficacy of the Coping With Depression Course for Adolescents (CWD-A) [138], the main components of which were social skills, pleasant activities, relaxation training, cognitive restructuring, conflict resolution, and relapse prevention; and (2) to test whether parental involvement enhanced treatment efficacy. Adolescents in CWD-A recovered significantly from depressive episodes and reduced their depressive symptoms, depressive cognitions, and anxiety significantly more than those in WL; improvement was maintained at a two-year follow-up. There was a trend in favor of the parent-involved group over the adolescent-only group; a significant difference between the two treatment groups was obtained on the depression, internalizing problems, and externalizing problems subscales of the Child Behavior Checklist [139] completed by the parents.

In a replication of the pioneering trial by Lewinsohn et al. [14], Clarke and colleagues [115] found that the recovery rate of treated adolescents was significantly higher than WL (66.7% versus 48.1%), an improvement confirmed by self-report of depression and clinician-rated global functioning. There were no differences, however, between the parent-and-adolescent group and adolescent alone group. Immediately after the post-treatment assessment, they introduced the novelty of randomly reassigning treated adolescents to three new conditions during the 24-month follow-up: (1) assessments every 4 months with booster sessions; (2) only assessments every 4 months; and (3) only assessments every 12 months. The booster sessions did not reduce the recurrence rate at follow-up, but they did appear to accelerate the recovery of adolescents who remained depressed at the end of the acute phase. A significant difference was also found for externalizing behaviors between adolescents who had received booster sessions and those who had only been assessed; participants who were evaluated quarterly outperformed those assessed annually on parental reports of depression and internalizing behaviors.

Rohde and colleagues [121] evaluated the effectiveness of the CWD-A course in depressed adolescents with comorbid conduct disorder, recruited from a juvenile justice department, compared with life skills tutoring (LST) that included current events review, life skills training (e.g., filling out a job application or renting an apartment), and academic tutoring. At the end of treatment, the rate of recovery from depression was significantly higher in CWD-A (39%) than in LST (19%), and depressive symptoms reduced more, according to adolescent self-report and clinician assessment. There also was an improvement in social functioning; the between-group difference in conduct disorder was not significant, however. The rate of recovery from depression was the same in both groups one year later (63%).

Clarke and colleagues [133] examined the feasibility of a brief CBT intervention in primary care with adolescents who had declined or discontinued treatment with antidepressants. In the first session of the acute phase, the adolescent and therapist together chose one of two program modules: cognitive restructuring or behavioral activation, which was implemented over four sessions. If, at the end of the first module, the adolescent was almost or completely recovered, they could end the intervention; if not, they were encouraged to continue with the other module. During the continuation phase, the adolescent had up to six optional treatment contacts. Recovery rates from depression in the follow-up period were significantly higher for CBT plus TAU than for TAU alone: 69.7% versus 43.4% (week 26), 79.8% versus 68.7% (week 52), 86.9% versus 75.8% (week 78), and 88.9% versus 78.8% (week 104). Median recovery time was 22.6 weeks for CBT plus TAU and 30 weeks for TAU alone. Compared with TAU alone, the reduction in clinician-rated and adolescent-reported depressive symptoms, reduction in dysfunctional thoughts and complaints, and improvements in global functioning and quality of life were superior for CBT plus TAU in the first year of follow-up, although not in the second year. The rate of psychiatric hospitalizations was also significantly lower for CBT plus TAU during the first year of follow-up.

In a replication of the initial trial of Reynolds and Coats [13] with depressed adolescents, Wood and colleagues [113] compared a program composed of Beck’s cognitive therapy [140], interpersonal problem solving, sleep hygiene, and pleasant activities, with another active treatment for depression and comorbid symptoms. Adolescents in the multicomponent program improved significantly more in depression and self-esteem and were more satisfied with the treatment than those who received only progressive relaxation training [141]. Both groups improved in anxiety, but not in antisocial behavior. At follow-up, the between-group difference narrowed, partly because of the high relapse rate in the CBT group and partly because the adolescents in the relaxation group continued to recover.

Brent and colleagues [114] tested the efficacy of a protocol that included Beck’s cognitive therapy [142], problem solving, emotion regulation, and social skills, with a mixed-modality of family therapy (FT) based on behavioral and systemic approaches, and with non-directive supportive therapy (NDST). Only 17.1% of the adolescents in the CBT group were still depressed at the end of treatment, whereas 32.3% and 42.4% of the FT and NDST groups, respectively, still had major depressive disorder (MDD); the difference was significant only with NDST. CBT showed significantly higher remission rates, defined as the absence of depression and a score < 9 on the Beck Depression Inventory [143] for at least three consecutive sessions, than the other treatments: 64.7% (CBT) versus 37.9% (FT) and 39.4% (NDST). CBT also outperformed the other treatments in the reduction in clinician-rated depressive symptomatology and in the credibility of the therapy. Finally, CBT was better than FT in adolescent-reported depressive symptoms.

Alavi and colleagues [127] evaluated the effectiveness of a CBT protocol for Suicide Prevention, developed by Stanley and colleagues [144], with the aim of decreasing suicidal ideation and hopelessness in depressed adolescents, who had had at least one suicide attempt in the previous quarter. In the initial phase of the treatment, which was conducted over three sessions, vulnerability factors and triggering events of suicidal crises were analyzed (i.e., chain analysis), coping strategies, and sources of support were provided (i.e., safety planning), the nature of suicide was explained together with the need to monitor potential lethal means (i.e., psychoeducation), the adolescent’s personal reasons for living were discussed (i.e., building hope), and a functional analysis of the problem was conducted (i.e., case conceptualization). The middle phase, sessions 4–9, included individual and family modules on behavioral activation, emotion regulation, cognitive restructuring, problem solving, and assertiveness. The three sessions of the final phase were devoted to relapse prevention. At the end of treatment, there were significant differences between youths in CBT versus those in the control condition on measures of depression, suicidal ideation, and hopelessness. Scores on these variables decreased between 54% and 77% in CBT, whereas they did not change in the condition that received only routine psychiatric intervention, including medication.

Charkhandeh and colleagues [132] compared CBT with the Reiki method of alternative medicine and WL. The goals of CBT were to modify distorted perceptions, learn problem-solving, acquire coping skills, and motivate behavior through enjoyable activities. At the conclusion of treatment, CBT had reduced depressive symptoms significantly more than either the Reiki method or WL.

Finally, we reviewed an RCT by Yang and colleagues [134] that compared active training in attentional bias modification (ABM) with placebo training. Although the treatment was very different from usual CBT, we included the trial because ABM is a specific target of cognitive therapy [142]. In the active condition, adolescents completed the neutral phase of ABM in eight sessions (320 trials per session), over two weeks, to shift their attention from sad words to neutral words. Nine weeks later, the positive phase of ABM took place in four sessions (480 trials per session), over two weeks, to redirect the attention from neutral words to positive words. The placebo condition used identical tasks but gave equal attention to sad and neutral words. After the neutral phase, a significantly higher proportion of adolescents in active ABM did not meet diagnostic criteria for MDD compared to placebo ABM (87% and 59%, respectively). The between-group difference was not maintained at the 7- or 11-week follow-ups, however. Active ABM also reduced clinician-rated depressive symptoms and attentional bias more than placebo ABM. At the 12-month follow-up, adolescents reported significantly fewer depressive and anxiety symptoms with active ABM as compared to placebo ABM.

#### 2.1.2. Cognitive–Behavioral Therapy Plus Medication

The Treatment for Adolescents with Depression Study (TADS) Team [122] conducted one of the largest adolescent depression RCTs to assess the effectiveness of CBT and/or fluoxetine versus pharmacological placebo. The starting dose of fluoxetine and placebo was 10 mg/day, which was increased to 20 mg/day at week 1 and, if necessary, up to 40 mg/day at week 8. The initial phase of CBT (sessions 1–6) included psychoeducation, mood monitoring, pleasant activities, social problem-solving, and cognitive restructuring. The middle phase (sessions 7–12) addressed adolescents’ social skill deficits, including problems with social engagements, communication, negotiation, compromise, and assertion. In the final phase (sessions 13–15), two sessions were held with the parents alone to provide psychoeducation, and one to three sessions were held with the parents and the adolescent, focused on the concerns of both parties. The combined treatment of CBT plus fluoxetine significantly reduced depressive symptoms as assessed by the clinician more than separate psychological and pharmacological treatments and/or the placebo. Treatment response at week 12, defined as being much or very much improved, was more positive in the combined treatment (71%) and fluoxetine alone (60.6%) than in CBT alone (43.2%) or the placebo (34.8%). Suicidal ideation decreased in all four groups, although the reduction was greater in the combined treatment. The authors concluded that the combination of CBT with fluoxetine offered the best balance between benefit and risk and adding CBT to medication enhanced safety.

In another combined treatment trial, Clarke and colleagues [123] asked whether the adjunct of a brief CBT intervention based on cognitive restructuring and/or behavioral activation would increase the effectiveness of selective serotonin reuptake inhibitor (SSRI) administered as TAU in a health maintenance organization. The acute phase CBT program was the same as previously described [133] and its duration varied depending on the adolescent’s degree of recovery. During the continuation phase of CBT, the therapist made six brief check-in telephone calls to the adolescent at one, two, three, five, seven and nine months after completing treatment. Adolescents in the SSRI condition could receive any other treatment from the health maintenance organization and/or other outside services. Over one year, Clarke et al. conducted four follow-up assessments, at weeks 6, 12, 26, and 52, and found that CBT had significant benefits for mental health status. There was a reduction in outpatient visits and in the number of days on all types of medication and a marginally significant difference in depressive symptoms and externalizing problems as reported by adolescents.

Melvin and colleagues [124] compared the combination of CBT and sertraline with CBT alone and sertraline alone. CBT consisted of the CWD-A adapted for individual use in therapy. The initial dose of sertraline was 25 mg/day. After the first week, the dose was adjusted depending on clinical response. If adverse effects occurred, the medication was halved during several days for one week, after which time it was increased again to 25 mg/day. Conversely, if no adverse effects were observed during the first week, the dose was doubled (i.e., 50 mg/day). Subsequently, the dose was increased by 25 mg to a maximum of 100 mg/day, depending on clinical response and tolerance. The most frequent adverse effects of sertraline were fatigue (31.1%), difficulty concentrating (24.4%), and insomnia (22.2%). The highest response to treatment was obtained in the CBT alone group (86%), and the lowest in the sertraline alone group (46%). The probability of occurrence of depression at post-treatment was significantly lower in the CBT alone group than in the sertraline alone group, whereas the combined treatment did not differ from either treatment alone. At the six-month follow-up, there were no differences between the groups. Thus, in contrast to the short-term findings of the TADS Team [122], the advantage of the combined treatment in this study was less evident.

In the Treatment of SSRI-Resistant Depression in Adolescents (TORDIA) trial, Brent and colleagues [126] compared the combination of CBT and an SSRI or venlafaxine with these drugs given separately to adolescents who had previously failed to respond to SSRI treatment. All participants’ parents, regardless of treatment group, received family psychoeducation about depression. The CBT included cognitive restructuring, behavioral activation, emotion regulation, social skills, and problem solving. The alternative SSRIs were paroxetine, citalopram, or fluoxetine, with a daily dose of 10 mg per day the first week, 20 mg per day in weeks 2–6, with an option to increase to 40 mg per day if there was no improvement. The starting dose of venlafaxine was 37.5 mg, which was increased by that same amount over the next three weeks (i.e., 75, 112.5, and 150 mg), with the possibility of reaching a maximum dose of 225 mg daily in the sixth week. CBT plus any antidepressant showed a significantly higher response rate than any medications alone; the effects of the various medications did not differ from each other. There were no differences between treatments on clinician-rated or adolescent-reported depressive symptoms, suicidal ideation, rate of related harm, or any other negative events. Adverse effects of medication, such as increased diastolic blood pressure and pulse or skin problems, were more frequent with venlafaxine.

### 2.2. Interpersonal Therapy

In the late 1990s, Mufson and colleagues [116] adapted the therapy developed by Klerman and colleagues [145], used to treat depression in adults, for the treatment of adolescents. The modification addressed common developmental issues in adolescence such as separation from parents, conflict with parental authority, exploration of dyadic interpersonal relationships, experience of the death of a family member or friend, and peer pressure. A specific area was added for single-parent families. The control condition was clinical monitoring, in which the therapist was limited to checking depressive symptoms and school attendance, assessing suicide risk, and practicing active listening, refraining from counseling, or providing skills training. Adolescents in both groups could contact the therapist if they felt worse (“call-me-if-you-need-me”). At the end of treatment in the intention-to-treat sample, 75% of the IPT adolescents had recovered from depression, scoring 6 or less on the Hamilton Rating Scale for Depression [146], whereas only 46% improved in the control condition. In addition, for youth in the IPT, there was a significantly greater reduction in depressive symptoms based on the clinician ratings and adolescent’s self-report, better overall social functioning with friends and with their partner, and greater interpersonal problem-solving skills, compared to youth in the control condition.

Rosselló and Bernal [117] evaluated the effectiveness of IPT and CBT compared with WL. The IPT used in this study was also an adaptation of the original model of Klerman and colleagues [145]. IPT was carried out in three blocks of four sessions each. The goals of sessions 1–4 were to elicit information about the adolescent’s depression, present the therapy, assess interpersonal relationships, identify core issues such as grief, interpersonal disputes, role transitions, and interpersonal deficits, establish a treatment plan, and explain what was expected of the adolescent in therapy. In sessions 5–8, they worked on selected interpersonal issues, monitoring depressive feelings, facilitating a positive therapeutic relationship, and preventing parental interference in the treatment. Sessions 9–12 discussed feelings related to separation, the treatment was reviewed, and the adolescents’ interpersonal competence was acknowledged. The CBT was based on Lewinsohn’s behavioral therapy [147,148], and Beck’s [142] and Ellis’ [149,150] cognitive therapy, the main components being cognitive restructuring, pleasant activities, and social skills. At both post-treatment and the three-month follow-up, adolescents that received treatment showed significantly fewer depressive symptoms than those who did not; there was no difference between IPT and CBT; and self-esteem and social adjustment improved with IPT compared with WL. No differences in these variables were found between IPT and CBT, nor between CBT and WL.

In 2004, the Mufson team [120] evaluated the effectiveness of IPT compared with TAU, from school-based health clinic clinicians. IPT was implemented as described in the manual by Weissman and colleagues [151], similar to those in the earlier trial by the Mufson team [116]. Most of the adolescents in the TAU group received individual psychotherapy–supportive counseling. At the conclusion of treatment, youth in the IPT showed significantly fewer clinician-rated depressive symptoms, better global and social functioning, greater clinical improvement, and a greater decrease in severity than TAU.

### 2.3. Family Therapy

The trial by Brent and colleagues [114], described earlier, included an FT condition. In the first phase of treatment, based on Alexander and Parsons’ [152] functional FT, the therapist addressed the concerns raised by the family and reframed issues to optimize treatment engagement and identify dysfunctional behavior patterns. During the second phase, adapted from Robin and Foster’s [153] problem-solving model, family members focused on communication, negotiation, and conflict resolution skills, as well as on the modification of inappropriate family relationship patterns. The protocol included psychoeducation on depression and the developmental and educational aspects of depression, and emphasized skills building and positive practice in sessions and at home. FT was significantly inferior to CBT and did not differ from NDST in the remission rate of depression. There also was a smaller reduction in depressive symptoms, as assessed by clinician ratings and adolescent report, for those in FT compared to CBT; the latter treatment seemed more credible to parents.

Diamond and colleagues [119] contrasted attachment-based FT with a minimal contact WL control. The therapy consisted of five tasks aimed at repairing attachment and promoting autonomy. The Relational Reframing Task replaced a focus on “fixing” the adolescent with one of improving family relationships and addressing parental criticism and hostility with the aim of reducing blame and increasing mutual respect. The Adolescent Alliance-Building Task sought to create a bond to increase motivation and treatment engagement, explore family conflicts that damage trust, and prepare to discuss these issues with parents. The Parent Alliance-Building Task explored the parent’s current stressors and history of attachment failures, as well as promoted an authoritative educational style. The Attachment Task was initiated when the adolescent expressed anger about core conflicts, usually related to betrayal, abandonment, or abuse, such that parental remorse promoted adolescent forgiveness. The Competence-Promoting Task fostered the adolescents’ relationships and success outside the home (e.g., school, peers, work); with attachment recovered, the family became the secure base from which the adolescents could explore their emerging autonomy. At post-treatment, 81% of the adolescents who had received FT did not meet criteria for MDD, in contrast to 47% on the WL. FT also achieved a significantly greater reduction in depressive and anxiety symptoms and family conflict. Of the 15 cases evaluated at six-months follow-up, 13 adolescents (87%) were free of depression.

In a Norwegian pilot study with a small sample, Israel and Diamond [128] implemented the protocol of the previous trial [119]. The control condition was individually dispensed treatment in outpatient clinics. FT reduced depressive symptoms significantly more than TAU when assessed by the clinician, but not by adolescent self-report.

Pool and colleagues [136] compared the BEST MOOD program [154], which incorporates elements of attachment theory, such as parental sensitivity, grief responses, and understanding of frightening or stressful family environments, with the Parenting Adolescents Support Training (PAST) program [155], a control condition similar to TAU. The first four BEST MOOD sessions were held with parents only and included strategies for engaging the child in the program, stress reduction techniques, psychoeducation on family and adolescent development, family unity, parent–child communication, and parental self-care. The adolescent and siblings joined the therapy in the final four sessions devoted to clarifying the family roles, addressing loss and trauma, improving communication patterns, implementing behavioral activation techniques with the adolescent, and promoting positive family rituals. Improvement in depressive symptoms was similar in the two groups at post-treatment, *d* = 0.83 and *d* = 0.80, BEST MOOD and PAST, respectively, and at three-months follow-up, *d* = 0.46 and *d* = 0.51, BEST MOOD and PAST, respectively. There was no difference between the programs in the other variables including emotional symptoms, alcohol consumption, efficacy, dependence, and self-criticism. The only positive effect of BEST MOOD was the greater reduction in symptoms of depression and stress in the parents at the three-month follow-up.

### 2.4. Psychoanalytic Therapy

The IMPACT trial [135] is the only one psychoanalytic therapy (PT) compared to CBT and a brief psychosocial intervention control condition. The short-term PT was based on close observation of the relationship which the adolescent established with the therapist. The therapist approached therapy as the task of understanding the adolescent’s feelings and life difficulties. The therapist was nonjudgmental and non-inquisitive and conveyed the value of self-understanding. The CBT focused on modifying behaviors and information processing biases through collaborative empiricism between the therapist and the adolescent. The brief psychosocial intervention control condition emphasized the importance of psychoeducation about depression, in addition to goal-focused and action-oriented interpersonal activities but did not include either self-understanding or cognitive change. There was no significant difference in adolescent-reported depressive symptoms between short-term PT and CBT, or between short-term PT and CBT versus the control condition, at either the conclusion of treatment (week 36) or at follow-up (week 86).

## 3. Status of Psychological Treatments for Depression in Adolescents

The first review that applied the Criteria for Empirically Supported Treatments [156] to evaluate the efficacy of psychological treatments [157] identified seven trials on adolescent depression, and considered CWD-A to be the only treatment that had achieved probable efficacious status, based on the two trials by the research team of Lewinsohn [14,25], which demonstrated the superiority of CBT over WL. This paucity of results is understandable, considering that when the review was published in 1998, barely three years had elapsed since the development of the classification criteria for evidence-based therapy.

A subsequent review published in 2008 [158] revealed the growing number of adolescent depression trials produced in one decade. They examined 18 new trials plus two published in 1996 [112,113], which had not been included in the earlier review, and concluded that CBT and IPT were well-established treatments, because there were at least two RCTs showing that they were superior to another treatment or to an active control condition.

A third 2016 review [159], collected on the website of the Society of Clinical Child and Adolescent Psychology, Division 53 of the American Psychological Association, on evidence-based mental health treatment for depression (Retrieved 13 April 2021 from https://effectivechildtherapy.org/concerns-symptoms-disorders/disorders/sadness-hopelessness-and-depression/), re-evaluated the 27 trials from the previous reviews—of which it eliminated six trials mainly for methodological reasons—additionally, it identified 14 new trials, confirming the well-established treatment status of CBT and IPT. The novelty was the addition of FT as a possibly efficacious treatment, because in the two trials by the research team of Diamond, FT was superior to an active control [128] and WL [119].

These reviews included treatment trials with clinical samples and prevention trials with subclinical samples. In the current review, we focused only on trials involving adolescents with a diagnosis of depression; that is, 18 of the previous trials, to which we added 9 more that were new or were not included [127,128,131,132,133,134,135,136,137] in the aforementioned reviews.

### 3.1. Cognitive–Behavioral Therapy

The body of research generated by CBT is far superior to that of other therapies. Six trials [13,14,19,23,25,114] in the 1998 review [157] included at least one CBT group, which was the only probably efficacious treatment. However, CBT is the first psychological treatment that, as early as the 1990s, achieved the status of a well-established treatment, because two trials conducted by independent research teams revealed that CBT was more effective than progressive relaxation training [113], FT, and NDST [114]. The discrepancy in evaluation is explained by the fact that the 1998 review did not include the study by Wood and colleagues [113]. The 2008 review [158] added fifteen trials, ten treatment [112,113,115,117,118,121,122,123,124,125] and five prevention [27,28,34,36,38], corroborating the well-established treatment status achieved by CBT in 1997. The 2016 review [159] updated the review of evidence-based treatments, removing five of the twenty-one previous trials and adding ten trials published between 2008 and 2014. They assigned well-established treatment status to individual CBT and to group CBT, possibly efficacious status to CBT using bibliotherapy, and experimental status to technology-assisted CBT.

We identified 96 CBT trials, of which we included 22 (Table 4) and excluded 74 (Table 2). Independent research teams have shown that CBT, applied individually, is superior to FT and NDST [114], relaxation training [113], the Reiki method [132], TAU [127,133], psychological placebo [134], and WL [117,132]. With respect to medication, CBT is more effective than sertraline [124]; when combined with fluoxetine, it is more effective than fluoxetine or a pharmacological placebo [122]; when combined with SSRI, CBT is more effective than an SSRI or venlafaxine (an SNRI) [123,126]; and when combined with venlafaxine, CBT is more effective than an SSRI or venlafaxine [126], making CBT a well-established treatment. Three trials by the same research team of Clarke, Rohde, Lewinsohn, and colleagues have shown that CBT, applied in a group setting, is superior to life skills tutoring [121] and WL [14,115], making it a probably efficacious treatment.

In recent years, there has been an increase in CBT programs via the internet (iCBT), computer (cCBT), or other devices such as cell phones [49,55,56,58,59,70,75,82,95,101], although these have been prevention trials with universal or indicated samples. One study [134] used a computer to present the stimulus words of an attention redirection task, although such attentional bias modification training is not really comparable to the usual technology-assisted CBT, of which we found only one trial with depressed adolescents [131]. Although this online intervention achieved greater improvement on all measures, including depressive symptoms, the difference with TAU was not significant. Similarly, CBT trials using bibliotherapy [27,47,83] were with indicated samples for prevention; thus, these modalities remain in the experimental phase.

### 3.2. Interpersonal Therapy

The only study included in the 1998 review [157] was the open trial by Mufson and colleagues [21] which involved 14 adolescents with depression, making IPT an experimental treatment up to that date. The 2008 review [158] identified three RCTs of IPT applied individually and one applied in a group setting. In the trial by Rosselló and Bernal [117], IPT was equivalent to CBT; CBT had previously achieved well-established status and was superior to WL.

In the trials by Mufson’s team, IPT was superior to clinical monitoring [116] and TAU [120]. In another trial by Mufson’s team, group IPT was more effective than school counseling [39]. Thus, individual IPT was considered to be a well-established treatment, because in at least two trials, conducted by different research teams, it proved to be as effective as a well-established treatment and superior to other treatments, whereas group IPT remains an experimental treatment.

The 2016 review [159] added two new trials conducted by the same two research teams. In the second trial by Rosselló’s team [46], IPT was shown to be a robust treatment, both individually and in groups, although the reduction in depressive symptoms and internalizing and externalizing behaviors, as well as the improvement in self-concept, were significantly lower than with CBT. In a new trial by Mufson’s team [53], IPT was superior to school counseling in reducing depressive symptoms and improving global functioning. Thus, individual IPT was judged to be a well-established treatment and group IPT was a probably efficacious treatment.

We identified ten IPT trials, of which seven were excluded because they were indicated to be prevention [39,40,53] or open-label trials [21,29], or involved preadolescents [73]. The exclusion of the most recent trial by Rosselló’s team [46] could be debatable; one out of three adolescents was selected by the clinical interviewer’s judgment based on the degree of impairment and two out of three met the criteria for MDD; this exclusion, however, does not affect the evaluation of IPT because it was significantly inferior to CBT. Of the remaining three trials, the first RCT by Mufson’s team [116] compared IPT with clinical monitoring, an “ethical WL condition” (p. 574), which consisted of three monthly 30 min sessions, with an option for a second session within the month. In contrast, IPT included 12 weekly 45 min sessions, i.e., psychological attention time was up to six times longer in the active treatment than in the control condition.

Despite these concerns, individual IPT maintains the status of a well-established treatment for adolescent depression because there are at least two RCTs, conducted by independent research teams, in which it was as effective as CBT [117], a treatment of proven efficacy, and superior to TAU [120]. In contrast, group IPT would be in the experimental phase because the trials of Mufson’s team [39,53] were indicated prevention, as was that of Bolton [40], who selected adolescent survivors of war and displaced persons in northern Uganda.

### 3.3. Family Therapy

In the 1998 review [157], only the trial by Brent and colleagues [114] applied an FT protocol, which was inferior to CBT and did not differ from NDST; therefore, it was an experimental treatment. The 2008 review [158] included two new trials. In the first trial, attachment-based FT [119] was shown to be superior to WL, but in the second trial, psychoeducation-based FT [37] did not differ significantly from TAU; thus, FT was rated to be an experimental treatment. The 2016 review [159] incorporated two new trials [51,69], plus a third one [43] in which they analyzed childhood depression. The two adolescent trials, one by Diamond and colleagues [51] and one by Rohde and colleagues [69], failed to find a significant reduction in depressive symptoms between FT and control conditions; thus, it did not change the possibly efficacious status.

In the current review, we excluded seven of the ten FT trials identified because the samples did not meet our inclusion criteria. In the trial by Sanford and colleagues [37], not all adolescents met criteria for MDD; the trial by Trowell and colleagues [43] involved children and adolescents with a mean age of less than 12 years; the trial by Connell and Dishion [45] included high-risk adolescents based on parent and teacher reports of emotional or behavioral problems. Participants in the trial by Diamond and colleagues [51] had suicidal ideation and a major depressive episode, anxiety, attention deficit hyperactivity disorder, oppositional defiant or conduct disorders. Horigian and colleagues [62] included participants who were in substance use treatment and had symptoms of depression and anxiety. Rohde and colleagues [69] studied adolescents with comorbid substance abuse and depressive disorders. In the sample of Diamond and colleagues [104], participants were adolescents with suicidal ideation and depressive or anxiety disorders. Although in the trials by Sanford and colleagues [37] and Rohde and colleagues [69], a significant percentage of the adolescents had a depressive disorder 71% and 72%, respectively, there was no significant difference in depression measures between FT and the groups with which they were compared. Disparate results were obtained in the four trials included in the current review. In the two trials of Diamond’s team, FT was superior to TAU [128] and WL [119], although in the other trials no difference was found with TAU [136] or supportive therapy, and FT was inferior to CBT [114]; thus, the status of FT as possibly efficacious did not change.

### 3.4. Psychoanalytic Therapy

Of the three reviews prior to ours, only the most recent one, in 2016 [159], included a trial of PT [43], that focused on individual psychodynamic psychotherapy. PT did not differ from system integrative FT. Therefore, PT was judged to be an experimental treatment for child and adolescent depression. We excluded the study by Trowell and colleagues [43] from the present review because it included children and adolescents aged 9 to 15 years old, and therefore aligned better with other trials of treating depression in children. We did identify one new RCT [135], in which short-term PT was no more effective than either CBT, or a brief psychosocial intervention. The absence of significant differences in outcome suggests that short-term PT is only experimental treatment at this time.

## 4. Looking to the Future with Hope

In the more than thirty years since the 1986 Reynolds and Coats trial [13], a wealth of data has been accumulated on the treatment of depression in adolescents, which justifies moderate optimism about the future. In this review, we used the conservative criteria for selecting trials that involved adolescents diagnosed with depression and focused primarily on improvements in depression. Table 5 presents the categorization of each treatment for depression in adolescents based on these criteria. Thus, our results differed somewhat from previous reviews [157,158,159]. We confirmed that individual CBT and individual IPT are well-established psychological treatments for depression in adolescents; CBT reached level 1 in 1997 [113,114] and IPT did so years later in 2004 [117,120]. In contrast, group CBT, and especially group IPT, were repositioned at lower levels in our review because group application was more frequent in preventive trials. Rosselló and colleagues [46] analyzed the efficacy of individual and group formats of both therapies in a sample of 112 Puerto Rican adolescents with depressive symptoms. Both formats reduced depressive symptoms from pre- to post-treatment, 9.98 units on the Children’s Depression Inventory (CDI) [160] with the individual format (*d* = 1.30) and 7.33 units (*d* = 1.22) with the group format. Similarly, in CBT, the reduction in depressive symptoms on the CDI was 10.58 units (*d* = 1.50), and for IPT it was 6.9 units (*d* = 1.33). CBT showed significantly greater improvement than IPT in depressive symptoms, self-concept, and internalizing and externalizing problems. In contrast, there was no statistical difference between the individual and group formats in any of the outcomes assessed. Clinical improvement, defined as a cut-off score of 12 on the CDI, was similar for both therapies: 62% in CBT and 57% in IPT.

O’Shea and colleagues [161] conducted an IPT-only trial with a sample of 39 adolescents, aged 13 to 19 years old, from the metropolitan area of Brisbane, Australia. Intention-to-treat analyses indicated a significant improvement in depression with both individual and group modalities, which was maintained at 12-months follow-up, although there was no significant difference between individual and group administration. These data suggest that group CBT and group IPT may work well, but the paucity of trials with clinical samples prevents according them superior status.

FT was upgraded from an experimental treatment to probably efficacious in 2013 [119,128], although this qualification should be scrutinized further in the future, because the only two trials of FT with a positive effect on adolescent depression were conducted with small samples by the same research team.

The present review is an important update due to the addition of the 2017 trial of short-term PT as an evidence-based therapy. In the IMPACT trial [135], conducted with the largest sample to date, at 36 weeks from baseline, short-term PT achieved an improvement of large magnitude (*d* = 1.40), which was maintained at 86 weeks follow-up (*d* = 1.77). Moreover, PT was not inferior to the improvement obtained with CBT at 36 weeks (*d* = 1.70) or at 86 weeks (*d* = 1.80). It will be important to replicate these findings in future randomized controlled trials, and demonstrate that the CBT and short-term PT, as conducted in this setting, are superior to an active control condition.

Thus, having treatments with demonstrated efficacy that work well for treating depression in adolescents, such as CBT and IPT, or that may be promising such as FT or short-term PT, is cause for cautious optimism. Several issues, however, limit total enthusiasm, but provide directions for continued progress in future research.

Firstly, in several trials, the improvement achieved with psychological treatments did not differ from that obtained with the control condition. An explanation for the absence of statistically significant differences could be, together with the high rate of response to placebo and spontaneous recovery from depression in adolescents, the nature of the control conditions used. For ethical reasons, WL is used less and less and, faced with the difficulty of finding a psychological placebo, TAU often is the control condition of choice. Over time, the quality of routine care provided in mental health clinics has improved considerably. Therefore, TAU should be considered more of an active treatment than a “no” or limited treatment control condition in RCTs on evidence-based therapy [162].

Secondly, a significant percentage of adolescents do not respond to treatment. Except for the trial by Esposito-Smythers and colleagues [137] with depressed adolescents with suicidal crises and concurrent risk factors, where flexible treatment was offered over one year, in the remaining 26 RCTs reviewed here, the response rate at post-treatment ranged from 27% to 29% in FT [128,136], to 86% to 89% in CBT and IPT [112,117,124,134], with an average of 60.9% across all studies. Why was a higher overall success rate not obtained? One salient predictor of poor response to treatment is comorbid anxiety disorders [163,164,165]. As such, transdiagnostic interventions are a possibly useful alternative. Ehrenreich-May and colleagues [91] implemented the Unified Protocol for the Treatment of Emotional Disorders in Adolescents (UP-A) with 51 adolescents with at least one primary diagnosis of generalized anxiety disorder (41.2%), social phobia (31.4%), major depression (21.6%) or other disorders (35.2%) and comorbid diagnoses. The core modules were: (a) knowledge of behaviors and emotions; (b) emotional awareness; (c) flexibility of thinking; (d) emotional exposure; and (e) maintenance of improvement. Additional modules included: (a) strengthening motivation; (b) coping with difficulties; and (c) parental education about the emotional adolescent. Transdiagnostic treatment significantly outperformed WL on all outcome measures at post-treatment. Sandín and colleagues [166] adapted the protocol for internet-based application (iUP-A), facilitating its implementation. Using the transdiagnostic protocol, Group Behavioral Activation Therapy (GBAT), based on live anti-avoidance exposure, Chu and colleagues [77] achieved a marginally significant reduction in the remission rate of the main diagnosis: 57.1% GBAT versus 28.6% WL.

Thirdly, specific treatment effects tend to fade over time [167]. Due to spontaneous recovery, most depressive episodes remit in 7–9 months in clinical samples [168], rather than worsening in treated adolescents. The course of depression is often chronic, with periods of remission; however, depression in adolescents is also recurrent between 46% and 63% [169]. Therefore, a relapse prevention component in the acute phase of treatment and booster sessions during the maintenance phase should be encouraged.

A fourth important question is to what degree should parents be involved in the treatment of depression in adolescents. Studies by Lewinsohn’s team [14,115] have not reliably shown the superiority of including parents. Spirito and colleagues [170] used a CBT protocol similar to that used in the TORDIA trial [126], with 24 dyads of adolescents with a suicidal history and a current major depressive episode and parents with a current or past major depressive episode. They hypothesized that treatment would be more effective if delivered jointly to the adolescent and parents than to the adolescent alone. The parent–adolescent group was significantly superior in reducing the depressive symptoms in both members of the dyad, especially the parents, at the end of the maintenance phase (24 weeks). There were no differences between groups at follow-up (48 weeks). Suicidal ideation was equally reduced in both groups during the acute and maintenance phases of treatment, and this improvement continued at follow-up. Interestingly, however, satisfaction with treatment was lower in the conjoint group; two adolescents were described by their therapist as oppositional and sessions with the family were needed to address parent–child conflict. Another adolescent reported abuse in the family and had to be hospitalized mid-treatment. In the absence of conclusive data and the higher cost of the involving parents in therapy, including parents in treatment of depression in adolescents needs further study and should be viewed with caution.

Additionally, the pharmacological treatment of adolescent depression is not without controversy. In 2006, Hammad and colleagues [171] conducted a meta-analysis with data from 4582 pediatric patients, who had participated in 24 trials—of which 16 dealt with MDD, including the multicenter TADS study—and reported that suicidality increased in pediatric patients treated with antidepressants by 1–3%. Consequently, the Food and Drug Administration (FDA) issued a “black box” warning about prescribing such medication to persons under 24 years of age [87]. Considering the concern raised and the response rate to antidepressants of approximately 60% [172], the first option for treating depression in adolescents will likely be psychological [173]. If the circumstances make combined treatment advisable, then the indicated drug is fluoxetine, in accordance with the indications of the U.S. FDA, the European Medicines Agency, and the Spanish Agency of Medicines and Health Products.

There is still a long way to go, but more than three decades of research have left us with two well-established and well-functioning psychological treatments for depression in adolescents: CBT and IPT. Other treatments, FT and short-term PT, require further RCTs to replicate the positive preliminary findings. Transdiagnostic protocols, delivery of therapy through information and communication technologies, and indicated prevention programs are currently expanding lines of research.

In conclusion, the first-line psychological treatments for depression in adolescents are individual CBT and individual IPT. The present review has been limited to adolescents with depressive disorders, excluding those with depressive symptoms; thus, it would be interesting to review the indicated prevention trials, especially considering that many of the excluded trials (Table 2) obtained positive results.

## Figures and Tables

**Table 1 ijerph-18-04600-t001:** Review criteria used for evidence base updates in the *Journal of Clinical Child and Adolescent Psychology.*

Methods criteria M.1. Group design: Study involved a randomized controlled design; M.2. Independent variable defined: Treatment manuals or logical equivalent were used for the treatment; M.3. Population clarified: Conducted with a population, treated for specified problems, for whom inclusion criteria were clearly delineated; M.4. Outcomes assessed: Reliable and valid outcome assessment measures gauging the problems targeted (at a minimum) were used; M.5. Analysis adequacy: Appropriate data analyses were used, and sample size was sufficient to detect expected effects.
Evidence criteria Level 1: Well-Established Treatments 1.1. Efficacy demonstrated for the treatment by showing the treatment to be either: 1.1.a. Statistically significantly superior to pill or psychological placebo or to another active treatment; OR 1.1.b. Equivalent (or not significantly different) to an already well-established treatment in experiments; AND 1.1.c. In at least two (2) independent research settings and by two (2) independent investigatory teams demonstrating efficacy; AND 1.2. All five (5) of the Methods criteria. Level 2: Probably Efficacious Treatments 2.1. There must be at least two good experiments showing that the treatment is superior (statistically significantly so) to a wait-list control group; OR 2.2. One (or more) experiments meeting the Well-Established Treatment level except for criterion 1.1.c. (i.e., Level 2 treatments will not involve independent investigatory teams); AND 2.3. All five (5) of the Methods criteria. Level 3: Possibly Efficacious Treatments 3.1. At least one good randomized controlled trial showing the treatment to be superior to a wait list or no-treatment control group; AND 3.2. All five (5) of the Methods criteria. OR 3.3. Two or more clinical studies showing the treatment to be efficacious, with two or more meeting the last four (of five) Methods criteria, but none being randomized controlled trials.
Level 4: Experimental Treatments 4.1. Not yet tested in a randomized controlled trial; OR 4.2. Tested in one or more clinical studies but not sufficient to meet level 3 criteria.
Level 5: Treatments of Questionable Efficacy 5.1. Tested in good group-design experiments and found to be inferior to other treatment group and/or wait-list control group, i.e., only evidence available from experimental studies suggests the treatment produces no beneficial effect.

**Table 2 ijerph-18-04600-t002:** Trials excluded from the current review.

Year	Author(s)	Treatment Condition(s)	Reason(s) for Exclusion
1986	Reynolds and Coats [13]	CBT Pleasant Activities + Cognitive Techniques CBT Progressive Relaxation WL	Indicated prevention: BDI ≥ 12, RADS ≥ 72, BID ≥ 20 (two adolescents BID = 18)
1990	Kahn et al. [18]	CBT CWD-A CBT Progressive Relaxation CBT Self–Modeling WL	Indicated prevention: BID ≥ 20
1991	Fine et al. [19]	CBT Social Skills ST Therapeutic and Mutual Support	No random assignment in the strictest sense
1994	Lewinsohn et al. [20]	CBT CWD-A Parents and Adolescents CBT CWD-A Adolescents WL	Not published in a peer-reviewed journal
1994	Mufson et al. [21]	IPT for Adolescents	Open clinical trial
1994	Reed [22]	CBT Structured Learning Therapy Attention-Placebo	Only holistic clinical judgments of improvement were used as outcome assessment measure
1995	Clarke et al. [23]	CBT Adolescent Coping with Stress Course TAU	Indicated prevention: CES-D ≥ 24
1996	Kroll et al. [24]	CBT Continuation Therapy	Historical Control Condition
1996	Lewinsohn et al. [25]	CBT CWD-A Parents and Adolescents CBT CWD-A Adolescents WL	The same trial as Lewinsohn et al. [20] (This trial was excluded from the count of studies)
1997	Feehan and Vostanis [26]	CBT Placebo	The same trial as Vostanis et al. [95]
1998	Ackerson et al. [27]	CBT Book “Feeling Good” Delayed-Treatment Condition	Indicated prevention: CDI ≥ 10, HRSD ≥ 10
2001	Clarke et al. [28]	CBT Adolescent Coping with Stress Course TAU	Selective prevention: Adolescent offspring of depressed parents
2001	Santor and Kusumakar [29]	IPT for Adolescents	Open clinical trial
2003	Puskar et al. [30]	CBT Teaching Kids to Cope TAU	Indicated prevention: RADS > 60
2003	Roberts et al. [31]	CBT PRP Usual Health Education	Preadolescents: M age = 11.9, range: 11–13. Indicated prevention: CDI = 10 (mean)
2004	Kerfoot et al. [32]	CBT brief TAU	Indicated prevention: MFQ ≥ 23
2004	Szigethy et al. [33]	CBT PASCET-Physical Illness	Open trial
2005	Asarnow et al. [34]	CBT Quality Improvement Intervention and/or Medication TAU	57.4% no diagnosis of depression (depressive symptoms)
2005	Jeong et al. [35]	Dance Movement Therapy WL	Indicated prevention: High depression score (Beckman Depression Inventory)
2005	Kowalenko et al. [36]	CBT Adolescents Coping with Emotions WL	Indicated prevention: CDI ≥ 18 Trial was randomized at the school level
2006	Sanford et al. [37]	FT Psychoeducation + TAU TAU	Diagnosis of depression in the last 6 months 28.9% no depression diagnosis at baseline
2006	Sheffield et al. [38]	CBT Universal Intervention CBT Indicated Intervention CBT Universal + Indicated Intervention No Intervention Condition	Universal and/or indicated prevention
2006	Young et al. [39]	IPT Adolescent Skills Training School Counseling	Indicated prevention: 16 ≤ CES-D ≤ 39 75.6% no depression diagnosis
2007	Bolton et al. [40]	IPT Group Creative Play Intervention WL	Indicated prevention: APAI ≥ 32 Adolescents with symptoms of depression, anxiety, and conduct problems
2007	Riggs et al. [41]	CBT + Placebo Fluoxetine + Placebo	Adolescents with a primary diagnosis of substance use disorder
2007	Szigethy et al. [42]	CBT PASCET-Physical Illness TAU	Indicated prevention: CDI ≥ 9
2007	Trowell et al. [43]	PT Focused Individual Psychodynamic Therapy FT Systems Integrative Familiar Therapy	Preadolescents: M age = 11.7
2008	Bahar et al. [44]	Problem-Based Group Therapy Occupational Therapy	Semi-experimental study. Selective prevention: Students, six months after an earthquake
2008	Connell and Dishion [45]	FT Adolescent Transitions Program School-As-Usual Control	Selective prevention
2008	Rosselló et al. [46]	CBT Individual CBT Group IPT Individual IPT Group	34% no diagnosis of depression (CDI ≥ 13)
2008	Stice et al. [47]	CBT Brief Adolescent Coping with Stress Course CBT Book “Feeling Good” Group Supportive–Expressive Intervention Assessment–Only Control Condition	MDD excluded (depressive symptoms)
2009	Garber et al. [48]	CBT TAU	Indicated and selective prevention: Adolescents with depressive symptoms, offspring of depressed parents
2009	O’Kearny et al. [49]	CBT MoodGYM Internet Program Usual curriculum	Universal prevention: All year 10 girls attending a single sex school
2009	Weisz et al. [50]	CBT PASCET TAU	Preadolescents: M age = 11.8, range: 8–15
2010	Diamond et al. [51]	FT Attachment-Based Family Therapy TAU Enhanced	Heterogeneous sample: 39.4% MDE, 7.6% Dd, 66.7%% AD, 57.6% ED
2010	Dobson et al. [52]	CBT Adolescent Coping with Stress Course Attention-Placebo “Let’s Talk”	MDD or Dd excluded (depressive symptoms)
2010	Young et al. [53]	IPT Adolescent Skills Training School counseling	Indicated prevention: 16 ≤ CESMD ≤ 39 82.5% no depression diagnosis
2011	Hayes et al. [54]	CBT Acceptance and Commitment Therapy TAU	26.4% no diagnosis of depression (out the clinical range for depression)
2011	Stallard et al. [55]	CBT CD-ROM “Think, Feel, Do” WL	Depressive or anxious symptoms
2012	Fleming et al. [56]	CBT SPARX Computerized Program WL	Indicated prevention: CDRS-R ≥ 30
2012	Gillham et al. [57]	CBT PRP Parents and Adolescents CBT PRP Adolescents School-As-Usual Control	Indicated prevention: CDI = 11.1 (mean)
2012	Kauer et al. [58]	CBT Mobile Phone Self-Monitoring Program Attention-Placebo	Youth: M age > 18, range 14–24 Indicated prevention: KPDS > 16
2012	Merry et al. [59]	CBT SPARX Computerized program TAU	Symptoms of mild to moderate depressive disorder
2012	Stallard et al. [60]	CBT Resourceful Adolescent Program Usual School Provision Attention-Placebo	Indicated prevention: SMFQ ≥ 5
2013	Carrion et al. [61]	CBT Behavioral, Cognitive and Insight Techniques WL	Selective prevention: Adolescents exposed to interpersonal violence
2013	Horigian et al. [62]	FT Brief Strategic Family Therapy TAU	Selective and indicated prevention
2013	Listug-Lunde et al. [63]	CBT CWD-A Culturally Modified Version TAU	Students with depressive symptoms
2013	McCarty et al. [64]	CBT Positive Thoughts and Action ST Individual Support Program	Indicated prevention: MFQ ≥ 14
2013	Nöel et al. [65]	CBT “Talk’n’ Time” WL	Selective prevention: Rural preadolescent girls
2013	Shirk et al. [66]	CBT Cognitive Restructuring, Relaxation, Behavioral Activation, Interpersonal Problem Solving	Open clinical trial
2013	Stikkelbroek et al. [67]	CBT Individual Program “D(o)epression Course” TAU	Project to study effectiveness of CBT for adolescent depression
2014	Chen et al. [68]	CBT Program “Children and Disaster: Teaching Recovery Techniques” ST Listening, reflection, and empathy techniques No Intervention Condition	Selective prevention: Adolescents who lost at least one parent in an earthquake
2014	Richardson et al. [16]	CBT Reaching Out to Adolescents in Distress and/or Medication TAU	39.6% no diagnosis of depression (depressive symptoms)
2014	Rohde et al. [69]	FT Followed by CBT CBT Followed by FT Coordinated FT and CBT	Selective and indicated prevention: Adolescents with comorbid depressive disorders (54% MDD, 18% Dd)
2014	Stasiak et al. [70]	CBT CD-ROM “The Journey” Attention-Placebo: Computerized Psychoeducation	Indicated prevention: CDRS-R ≥ 30, RADS-2 ≥ 76
2014	Wijnhoven et al. [71]	CBT PRP WL	Indicated prevention: CDI ≥ 16
2015	Compas et al. [72]	CBT Family Group Written Information	Selective prevention: Preadolescents (M age = 11.5) of parents with depression
2015	Dietz et al. [73]	IPT Family-Based CCT Child-Centered Therapy (Rogerian model)	Preadolescents: M age = 10.8, range: 7–12
2015	Rickhi et al. [74]	Spirituality Informed e-Mental Health Intervention WL	M age > 18, range: 13–24 Inclusion criteria: Suspicion they might be suffering from depression
2015	Smith et al. [75]	CBT Stressbusters Computerized Program WL	Indicated prevention: MFQ ≥ 20
2016	Bella-Awusah et al. [76]	CBT WL	Indicated prevention: BDI-II ≥ 18 Trial was randomized at the school level
2016	Chu et al. [77]	CBT Transdiagnostic Behavioral Activation WL	Principal diagnosis: 17.1% depression, 82.9% anxiety disorder
2016	De Voogd et al. [78]	Active Online Emotional Working Memory Training Placebo Online Emotional Working Memory Training	Symptoms of anxiety and depression
2016	Fristad et al. [79]	Omega-3 Polyunsaturated Fatty Acids (Ω3) Psychoeducational Psychotherapy (PEP) Ω3 + PEP	Preadolescents: M age = 11.6, range: 7–14
2016	Gaete et al. [80]	CBT Normal teaching activities at school	Indicated prevention: BDI-II ≥ 10 (boys), BDI-II ≥ 15 (girls)
2016	Goossens et al. [81]	CBT Preventure Program No Intervention Condition	Selective prevention: Adolescents who drink alcohol
2016	Ip el al. [82]	CBT Grasp the Opportunity Website Attention control: An Anti-Smoking Website	Indicated prevention: 11 < CES-D < 41
2016	Jacob and de Guzman [83]	CBT Based-Bibliotherapy Intervention No Intervention Condition	Indicated prevention: BDI-II > 14, AADS > 61, KADS-11 > 12
2016	Jacobs et al. [84]	CBT Rumination-Focused Assessment Only Control	Adolescents at risk for depressive relapse
2016	McCauley et al. [85]	CBT Behavior Activation EBP-D	Diagnosis of depression or CDRS-R ≥ 45
2016	Poppelaars et al. [86]	CBT PRP (Dutch version: Op Volle Kracht) CBT SPARX Computerized Program CBT PRP + SPARX Monitoring Control Condition	Indicated prevention: RADS-2 ≥ 59
2016	Rice et al. [87]	Omega-3 Polyunsaturated Fatty Acids + CBT Cognitive Behavioral Case Management Paraffin Oil Placebo + CBT Cognitive Behavioral Case Management	Project “The Fish Oil Youth Depression Study (YoDA-F)”. Young: Age range 15–25
2016	Schleider and Weisz [88]	Single-Session Teaching Growth Personality Mindsets ST	Symptoms of anxiety and depression: RCADS-P T-score ≥ 60
2016	Takagaki et al. [89]	CBT Behavioral Activation No Intervention Condition	Indicated prevention: BDI-II ≥ 10 M age = 18.2; range: 18–19
2017	Barry et al. [90]	CBT Group Coaching Intervention No Intervention Condition	Indicated prevention: CES-DC ≥ 15 Not published in a peer-reviewed journal
2017	Ehrenreich-May et al. [91]	CBT UP-A WL	Principal diagnosis: 21.6% MDD, 3.9% Dd, 2.9% DD NOS, 41.2% GAD, 31.4% SP
2017	Ranney et al. [92]	TBI Motivational Interviewing CBI Motivational Interviewing TAU Enhanced	Indicated (CES-D-10 = 13.2 mean) and selective prevention: Adolescents presenting to Emergency Department at Level 1
2017	Shomaker et al. [93]	CBT Mindfulness: “Learning to BREATHE” CBT Blues Program	Indicated (CES-D ≥ 16) and selective prevention: Adolescent girls at risk for type 2 diabetes
2017	Tompson et al. [94]	CBT Family-Focused Treatment for Child Depression ST Individual	Preadolescents: M age = 10.8, range: 7–14
2017	Wright et al. [95]	CBT Stressbusters Computerized Program Attention Control: Accessing Low Mood Self-Help Websites	Indicated prevention: MFQ ≥ 20
2018	Bai et al. [96]	CBT Behavioral Health Intervention TAU Enhanced	48% no diagnosis of depression (CES-D = 20.1, mean). Adolescents with health risk behaviors
2018	Díaz-González et al. [97]	CBT Mindfulness-Based Stress Reduction TAU	Adolescents attending Mental Health Services: 11.3% MDD, 21.3% AD, 67.5% Other disorders
2018	Högberg and Hällström [98]	CBT Systematised Mood-Regulation TAU	Symptoms of depression tested with SMFQ
2018	Jensen-Doss et al. [99]	CBT UP-A + YOQ TAU + YOQ TAU	Adolescents with significant symptoms of anxiety or depression: CSR ≥ 4
2018	Singhal et al. [100]	CBT Coping Skills Program Interactive Psychoeducation	Indicated prevention: 14 ≤ CDI ≤ 24 Trial was randomized at the school level
2018	Topooco et al. [101]	CBT Internet-Based Attention-Placebo	24.3% no diagnosis of depression (depressive symptoms only)
2019	Brown et al. [102]	CBT DISCOVER ‘How to Handle Stress” WL	27.33% depression ‘cases’ 48.7% anxiety ‘cases’
2019	Davey et al. [103]	CBT + Fluoxetine CBT + Pills Placebo	M age = 19.6; range: 15–25
2019	Diamond et al. [104]	FT Attachment-Based ST Nondirective	Indicated prevention: BDI-II > 20 41.2% MDD, 3.9% Dd, 46.9% AD
2019	Grupp-Phelan et al. [105]	STAT-ED Motivational Interviewing TAU Enhanced	Selective prevention: Suicidal adolescents (ASQ)
2019	Idsoe et al. [106]	CBT Adolescent Coping with Depression Course TAU	Indicated prevention: CES-D ≥ 28
2019	Sánchez-Hernández et al. [107]	CBT Smile Program No Intervention condition	Indicated prevention: CDI > 10
2020	García-Escalera et al. [108]	CBT Internet UP-A WL	Universal prevention
2020	Osborn, Rodriguez et al. [109]	SI Single-Session Digital Intervention Digital Study Skills Condition	Universal prevention
2020	Osborn, Venturo-Conerly et al. [110]	Shamiri Intervention: Growth-Mindset Module + Gratitude Module + Value Affirmations Module Study Skills Condition	Indicated prevention: PHQ-8 ≥ 28 (depression), GAD-7 ≥ 10 (anxiety)
2020	Osborn, Wasil et al. [111]	Shamiri Intervention: Growth-Mindset Module + Gratitude Module + Value Affirmations Module Study Skills Condition	Indicated prevention: 37.3% adolescents reported moderately severe-to-severe depressive symptoms, 92.2% moderate-to-severe anxiety symptoms

AADS = Asian Adolescent Depression Scale; AD = Anxiety Disorder; APAI = Acholi Psychosocial Assessment Instrument; ASQ = Ask Suicide Screening Questions; BDI = Beck Depression Inventory; BID = Bellevue Index of Depression; CBI = Computer-delivered Brief Intervention; CBT = Cognitive Behavioral Therapy; CDI = Children’s Depression Inventory; CDRS-R = Children’s Depression Rating Scale-Revised; CES-D(C) = Center for Epidemiological Studies-Depression Scale (for Children); CSR = Clinical Severity Rating; CWD-A = Coping with Depression Course for Adolescents; DD = Depressive Disorder; Dd = Dysthymic Disorder; EBP-D = Evidence-Based Practice for Depression; ED = Externalizing Disorder; FT = Family Therapy; GAD(-7) = Generalized Anxiety Disorder (Screener-7); HRSD = Hamilton Rating Scale for Depression; IPT = Interpersonal Therapy; KADS-11 = Kutcher Adolescent Depression Scale; KPDS = Kessler Psychological Distress Scale; MDD = Major Depression Disorder; MDE = Major Depression Episode; NOS = Not Otherwise Specified; PASCET = Primary and Secondary Control Enhancement Training; PEPT = Psychoeducational Psychotherapy; PHQ-8 = Patient Health Questionnaire-8-item version; PT = Psychodynamic Therapy; PRP = Penn Resiliency Program; RADS = Reynolds Adolescent Depression Scale; RCADS-P = Revised Child Anxiety and Depression Scale-Parent Form; SI = Shamiri Intervention; (S)MFQ = (Short) Mood and Feelings Questionnaire; SPARX = Smart, Positive, Active, Realistic, X-factor thoughts; ST = Supportive Therapy; STAT-ED = Suicidal Teens Accessing Treatment After an Emergency Department Visit; TAU = Treatment As Usual; TBI = Therapist-delivered Brief Intervention; UP-A = Unified Protocol for the Treatment of Emotional Disorders in Adolescents; WL = Wait-list Condition; YOQ = Youth Outcomes Questionnaire.

**Table 3 ijerph-18-04600-t003:** Sociodemographic and clinical characteristics of the participants in trials included in the current review.

Year	Authors	N	Mean Age (Range)	Gender Female	Family Demographics	Ethnicity	Diagnosis	Suicidality	Comorbidity
1990	Lewinsohn et al. [14]	59	16.2 (14–18)	61%	40.7% Both parents 52.5% One parent 6.8% Neither parent		49% MDD 7% mDD 44% IDD	40% HSA	
1996	Vostanis et al. [112]	57	12.7 (8–17)	56.1%	50.9% Both parents 29.8% Single parent 7% Adoptive parents 12.3% Others	87.7% White 8.8% Asian 3.5% Black	29.8% MDD 54.4% mDD 15.8 Dd		45.6% OAD or SAD 19.3% ODD or CD
1996	Wood et al. [113]	53 (48) ^a^	14.2 (9–17)	68.8%			91.5% MDD 27% EDD		56% OAD 23% CD
1997	Brent et al. [114]	107	15.6 (13–18)	75.7%	57% Both parents	83.2 White	77.6% MDD 22.4% MDD + Dd	36.4% CSI 23.4% HSA	31.8% AD 20.6% DBD
1999	Clarke et al. [115]	123 (96) ^a^	16.2 (14–18)	70.8%	43.8% Both parents		76% MDD 12.5% Dd 11.5% MDD + Dd		23.6% AD
1999	Mufson et al. [116]	48	15.8 (12–18)	70.9%	68.8% One parent	70.8% Hispanic	79% MDD 21% MDD + Dd	42.5% CSI 27.5% HSA	88% AD
1999	Rosselló and Bernal [117]	71	14.7 (13–18)	54%			24% MDD 76% MDD + Dd		
2002	Clarke et al. [118]	88	15.3 (13–18)	69.3%	82.7% Parent female 4.6% Parent minority 77% Parents married 23.3% Parent college graduate 74.7% Employed	9.1% Minority	93.2% MDD 3.4% Dd 1.1% NOS BD		22.7% PTSD 18.2% ODD 4.5% SA 2.3% NOS Ed
2002	Diamond et al. [119]	32	14.9 (13–17)	78%	80% One parent 69% < USD 30,000 annual income 34% ≤ USD 20,000 annual income	69% African American 31% White	100% MDD		
2004	Mufson et al. [120]	63	15.1 (12–18)	84.1%	69.8% One parent	71.4% Hispanic	50.8% MDD 17.5% Dd 14.3% ADDM 11.1% NOS DD 6.3% dD	33.3% CSI 11.1% HSA	
2004	Rohde et al. [121]	93	15.1 (13–17)	48.4%	15.1% Both biological parents 14.8% Parent with bachelor’s degree or higher	80.6% White	100% MDD	39.8% HSA	100% CD
2004	TADS [122]	439	14.6 (12–17)	54.4%	41% One parent USD 50,000–74,000 modal family income	73.8% White 12.5% Black 8.9% Hispanic	100% MDD 10.5% Dd		27.4% AD, 23.5% DB, 13.7% ADHD, 4.3% Others
2005	Clarke et al. [123]	152	15.3 (12–18)	77.6%		13.8% Minority	100% MDD		
2006	Melvin et al. [124]	73	15.3 (12–18)	65.8%	58.5% Secondary school 41.5% Tertiary school		60.3% MDD 23.3% Dd 16.4% NOS DD		37% AD 26% PCRP 8.2% CD/ODD 15% Others
2007	Goodyer et al. [125]	208	14 (11–17)	74%			100% MDD 0.5% Dd		44.2% SP, 38% OCD, 37.5% PTSD, 31.2% AP, 28.4% SAD, 22.6% sP
2008	Brent et al. [126]	334	15.9 (12–18)	69.8%	USD 61,000 median family income	82.9% White 17.1% Other Ethnicity	100% MDD 29.3% Dd	23.7% HSA	36.4% AD 15.6% ADHD 9.9% ODD/CD
2013	Alavi et al. [127]	30	16.1 (12–18)	90%			100% MDD	100% HSA	
2013	Israel and Diamond [128]	20	15.6 (13–17)	55%			100% MDD		85% ID 55% ED 40% Ap
2014	Shirk et al. [129]	43	15.5 (13–17)	83.7%		49% Non-Hispanic Caucasian 38% African American 33% Hispanic	81.4% MDD 7% Dd 11.6% NOS DD		46% PTSD 14% SA
2014	Szigethy et al. [130]	217	14.3 (9–17)	51%		89.4% White 10.6% Black	63.1% MDD 36.9% mDD		74.2% Cd 25.8% UC
2015	Kobak et al. [131]	65	15.4 (12–17)	66.2%		41.5% Caucasian 36.9% African American 4.6% American Indian 1.5% Asian, 7.7% Biracial, 7.7% Others	47.7% MDD 30.8% PDD 4.6% MDD and PDD 7.7% NOS DD		
2016	Charkhandeh et al. [132]	188	(12–17) 12.8% 12–13 36.7% 14–15 50.5% 16–17	53.7%	86.2% Both parents 8.5% Only mother 3.2% Only father 2.1% None 30.9% > USD 800 68.6% < USD 800		100% MDD		
2016	Clarke et al. [133]	212	14.7 (12–18)	68.4%	$64,073 average family income	16% Hispanic 11.8% Minority	100% MDD		
2016	Yang et al. [134]	45	15 (12–18)	55.6%		100% Chinese population	100% DD	24.4% CSI/HSA	
2017	Goodyer et al. [135]	470 (465) ^b^	15 (11–17)	74.8%		84.5% White	100% MDD	34.4% HSA	12% ODD/CD
2018	Poole et al. [136]	64	15.2 (12–18)	73.4%	37.5% Married 37.5% Divorced 17.9% Single 19% USD 0–20,000 36% USD 20,000–50,000 21% USD 50,000–80,000 24% > USD 80,000		100% MDD, mDD or Dd		
2019	Esposito-Smythers et al. [137]	147	14.9 (12–18)	76.2%		85.5% White 2.1% Black/African American 2.8% Asian/Pacific Islander 9.7% Multiracial	89.1% MDD 10.9 Dd or NOS DD	65.5% HSA	39.6% GAD, 26.6% ADHD, 22.2% SAD, 18.8% ODD/CD, 18.3 PTSD

^a,b^ Data from ^a^ completers or from ^b^ intent-to-treat. AD = Anxiety Disorder; ADDM = Adjustment Disorder with Depressed Mood; ADHD = Attention Deficit Hyperactivity Disorder; AP = Agoraphobia; Ap = Attention Problems; BD = Bipolar Disorder; BN = Bulimia Nervosa; CD = Conduct Disorder; Cd = Crohn’s Disease; CSI = Current Suicidal Ideation; DB(D) = Disruptive Behavior (Disorder); DD = Depressive Disorder; Dd = Dysthymic Disorder; dD = Double Depression; DS = Depressive Symptomatology; ED = Externalizing Disorder; Ed = Eating Disorder; EDD = Endogenous Depression Disorder (RCD); GAD = Generalized Anxiety Disorder; HAS = History of Suicide Attempt; ID = Internalizing Disorder; IDD = Intermittent Depressive Disorder (RDC); MDD = Major Depression Disorder (DSM); mDD = Minor Depression Disorder (RDC); MDE = Major Depression Episode; NOS = Not Otherwise Specified; OAD = Overanxious Disorder; OCD = Obsessive Compulsive Disorder; ODD = Oppositional Defiant Disorder; PCRP = Parent–Child Relational Problem; PDD = Persistent Depressive Disorder; PTSD = Posttraumatic Stress Disorder; SA(D) = Substance Abuse (Disorder); SAD = Separation Anxiety Disorder; SP = Social Phobia; sP = Specific Phobia; SU(D) = Substance Use (Disorder); UC = Ulcerative Colitis.

**Table 4 ijerph-18-04600-t004:** Adolescent depression treatment outcome of trials included in the current review.

Year	Authors	Treatment Conditions	Sessions	Measures (Sources)	Posttreatment			Follow-Up	
					Improvement	Effect Size	Response Rate		
1990	Lewinsohn et al. [14]	CBT Parent and Adolescent CBT Adolescent WL	14 two-hour group over 7 weeks	CES-D (A) BDI (A) CBCL-D (P)	CBT (PA) ≥ CBT (A) > WL	CES-D: 1.51 PA, 1.18 A BDI: 1.48 PA, 0.94 A CBCL-D: 1.35 PA, -0.13 A	Loss of Diagnosis CBT (PA): 47.6% CBT (A): 42.9% WT: 5.3%	24 months Improvement was maintained	
1996	Vostanis et al. [112]	CBT PL	9 individual biweekly	MFQ (A, P)	CBT = PL	MFQ: 0.05 A, 0.51 P	Loss of Diagnosis CBT: 87% PL: 75%	9 months (recovered) 86% CBT, 75% PL 24 months 74.1% CBT, 85% PL	
1996	Wood et al. [113]	CBT RT Progressive Relaxation	8 individual weekly	MFQ-C (A, P)	CBT > RT	MFQ: N/A A, 0.41 P	MFQ-C Clinical Significance CBT: 75% RT: 33%	3 months: *d* = −0.06 6 months: *d* = 0.14	
1997	Brent et al. [114]	CBT FT Systemic and Behavioral NDST	12–16 individual over 12–16 weeks	K-SADS (C) BDI (A)	CBT > FT = NDST	K-SADS: 0.45 CBT, 0.14 FT BDI: 0.41 CBT, 0.07 FT	Loss of Diagnosis + BDI < 9 (3 Sessions) CBT: 82.9% FT: 67.7% NDST: 57.6%	12 months (recovered) 96.7% (rapid responders), 68.7% (initial non-responders) 24 months (recovered) No between-group differences	
1999	Clarke et al. [115]	CBT Parent and Adolescent CBT Adolescent WL	16 two-hour group over 8 weeks	HRSD (C) BDI (A) CBCL-D (P)	CBT (PA) = CBT (A) > WL	HRSD: 0.14 PA, 0.52 A BDI: 0.24 PA, 0.58 A CBCL-D: −0.43 PA, −0.47 A	Loss of Diagnosis CBT (PA): 68.8% CBT (A): 64.9% WT: 48.1%	12 months (recovered) 100% booster, 50% assessment 24 months (recovered) 100% booster, 90% assessment	
1999	Mufson et al. [116]	IPT CM	12 45 min individual weekly	HRSD (C) BDI (A)	IPT > CM	HRSD: 0.66 BDI: 0.66	HRSD ≤ 6 IPT: 75% CM: 46%	Not reported	
1999	Rosselló and Bernal [117]	CBT IPT WL	12 one-hour individual weekly	CDI (A)	CBT = IPT > WL	CDI: 0.35 CBT, 0.76 IPT	CDI < 12 CBT: 76% IPT: 89% WL: 66%	3 months CBT = IPT	
2002	Clarke et al. [118]	CBT + TAU TAU	16 two-hour group over 8 weeks	HRSD (C) CES-D (A) CBCL-D (P)	CBT = TAU	HRSD: 0.10 CES-D: 0.20 CBCL-D: −0.24	Few or no Depressive Symptoms ≥ 8 Weeks CBT: 31.6% TAU: 29.8%	12 months (recovered) 71.1% CBT, 82.1% TAU 89.5% CBT, 92.3% TAU	
2002	Diamond et al. [119]	FT Attachment-based WL	12 60–90 min family group weekly	HRSD (C) BDI (A)	FT > WL	HRSD: 0.64 BDI: 0.77	Loss of Diagnosis FT: 81% WL: 47%	6 months (recovered) 87% FT	
2004	Mufson et al. [120]	IPT TAU	12 35 min individual over 16 weeks	HRSD (C) BDI (A)	IPT > TAU	HRSD: 0.50 BDI: 0.37	HRSD ≤ 6 IPT: 50% TAU: 34%	Not reported	
2004	Rohde et al. [121]	CBT LST	16 two-hour group over 8 weeks	HRSD (C) BDI-II (A)	CBT > LST	HRSD: 0.39 BDI-II: 0.17	Loss of Diagnosis CBT: 38.6% LST: 19.1%	6 months (recovered) 54% CBT, 60% LST 12 months (recovered) 63% CBT, 63% LST	
2004	TADS [122]	CBT Fluoxetine CBT + Fluoxetine Pill Placebo	15 50–60 min individual over 12 weeks	CDRS-R (C) RADS (A)	CBT + FL > FL > CBT = PL	CDRS-R: −0.03 CBT, 0.68 FL, 0.98 CBT+FL RADS: −0.10 CBL, 0.50 FL, 0.82	CGI ≤ 2 CBT + FL: 71% FL: 60.6% CBT: 43.2% PL: 34.8%	Not reported	
2005	Clarke et al. [123]	CBT Brief + SSRI (TAU) SSRI (TAU)	5–9 one-hour individual	HRSD (C) CES-D (A) CBCL-D (P)	CBT + SSRI ≥ SSRI	HRSD: 0.05 CES-D: 0.17 CBCL-D: 0.09	No CMDE CBT + SSRI: 77% SSRI: 72.1%	12 months (recovered) 80.3% CBT + SSRI, 94.2% SSRI	
2006	Melvin et al. [124]	CBT Sertraline CBT + Sertraline	12 50 min individual weekly	RADS (A)	CBT > SER CBT + SER = CBT CBT + SER = SER	CBT vs. SER: 0.42 CBT vs. CBT + SER: 0.33 SER vs. CBT + SER: −0.07	Full Remission MDD CBT: 86% SER: 46%	6 months (recovered) CBT = SER = CBT + SER	
2007	Goodyer et al. [125]	CBT + SSRI + TAU SSRI + TAU	19 55 min individual over 28 weeks	CDRS-R (C) MFQ (A)	CBT + SSRI + TAU = SSRI + TAU	CDRS-R: −0.11 MFQ: −0.22	CGI ≤ 2 CBT + SSRI + TAU: 53.1% SSRI + TAU: 60.7%	Not reported	
2008	Brent et al. [126]	CBT + SSRI SSRI CBT + Venlafaxine Venlafaxine	12 60–90 min individual weekly	CDRS-R (C) BDI (A)	CBT + SSRI or Venlafaxine > SSRI = Venlafaxine	CBT vs. Medication: 0.09 CDRS-R, −0.05 BDI CBT vs. SSRI: 0.07 CDRS-R, 0.04 BDI CBT vs. Venlafaxine: 0.01 CDRS-R, −0.10 BD	CGI ≤ 2 CBT: 59% Medication: 47.6%	15 months (recovered) 89% without MDD	
2013	Alavi et al. [127]	CBT + TAU TAU	12 individual weekly	BDI (A)	CBT > TAU	BDI: 2.88	BDI Decrement CBT: 54% TAU: −0.1%	Not reported	
2013	Israel and Diamond [128]	FT Attachment-based TAU	12–16 family group weekly	HRSD (C) BDI-II (A)	FT > TAU	HRSD: 1.10 BID-II: 0.80	HRSD < 9 FT: 27% TAU: 11%	Not reported	
2014	Shirk et al. [129]	CBT Interpersonal Trauma TAU	12 individual weekly	BDI-II (A)	CBT = TAU	BDI-II: −0.95	Loss of Diagnosis CBT: 50.0% TAU: 48.0%	1 month BDI-II: *d* = −2.98	
2014	Szigethy et al. [130]	CBT PASCET NDST Supportive listening	12 45 min individual weekly	CDRS-R (C)	CBT = NDST	CDRS-R: 1.31 CBT, 1.30 NDST	CDRS-R ≤ 28 CBT: 67.7% NDST: 63.2%	Not reported	
2015	Kobak et al. [131]	CBT Technology-assisted TAU	12 weeks	QIDS (A)	CBT = TAU	QIDS: 0.08	CGI ≤ 2 CBT: 71.4% TAU: 60%	Not reported	
2016	Charkhandeh et al. [132]	CBT Reiki WL	24 90 min individual over 12 weeks	CDI (A)	CBT > Reiki > WL	CBT vs. Reiki: 1.11 CBT vs. WL: 2.03 Reiki vs. WL: 0.76	CDI Decrement CBT: 32.4% Reiki: 12.2% WL: 0%	Not reported	
2016	Clarke et al. [133]	CBT Brief Individual + TAU TAU	5–9 individual	CDRS-R (C) CES-D (A)	CBT > TAU	CDRS-R: 0.60 CES-D: 0.37	Loss of Diagnosis CBT: 31.3% TAU: 12.1%	24 months CBT: 88.9% TAU: 78.8%	
2016	Yang et al. [134]	CBT ABM Placebo ABM	8 individual over 2 weeks + 4 individual over 2 weeks	K-SADS (C) HRSD (C) CES-D (A)	ABM > PL	K-SADS: 0.60 HRSD: 0.63 CES-D: 0.07	Loss of Diagnosis ABM: 87% PL: 59%	12 months CES-D: *d* = 0.94	
2017	Goodyer et al. [135]	CBT PT Short-term BPI	20 over 30 weeks 28 over 30 weeks 12 over 20 weeks	MFQ (A)	CBT = PT CBT and PT = BPI	CBT vs. PT: 0.16 CBT vs. BPI: 0.40 PT vs. BPI: 0.25	Loss of Diagnosis CBT: 69% PT: 64% BPI: 56%	12 months CBT: 74% PT: 73% BPI: 71%	20 months CBT: 75% PT: 85% BPI: 73%
2018	Poole et al. [136]	FT Best Mood Program TAU PAST Program	8 two-hour family group	SMFQ (A)	FT = TAU	SMFQ: 0.07	SMFQ Decrement FT: 29.6% TAU: 23.8%	3 months *d* = −1.02	
2019	Esposito-Smythers et al. [137]	CBT Family-focused TAU Enhanced	0–6 months: weekly (A), biweekly (P) 6–9 months: biweekly (A), biweekly-monthly (P) 9–12 months: monthly (A, P)	K-SADS (C) CDI-2 (A)	CBT = TAU	CDI-2: 0.06	Loss of Diagnosis CBT: 79% TAU: 86.4%	6 months CBT: 72.6% TAU: 87.5% CDI-2: *d* = −0.56	

A = Adolescent; ABM = Attention Bias Modification; BA = Behavioral Activation; BDI = Beck Depression Inventory; BPI = Brief Psychosocial Intervention; C = Clinician; CBCL-D: Child Behavior Checklist-Depression; CBT = Cognitive Behavioral Therapy; CDI = Children’s Depression Inventory; CDRS-R = Children’s Depression Rating Scale-Revised; CES-D = Center for Epidemiological Studies-Depression Scale; CGI = Clinical Global Impressions; CM = Clinical Monitoring; CMDE = Current Major Depressive Episode; EBP-D = Evidence-Based Practice for Depression; FL = Fluoxetine; FT = Family Therapy; HRSD = Hamilton Rating Scale for Depression; IPT = Interpersonal Therapy; K-SADS = Kiddie School Age Schedule for Affective Disorders and Schizophrenia; LST = Life Skills Tutoring; MDD = Major Depression Disorder; NDST = Non-Directive Supportive Therapy; P = Parents; PASCET = Primary and Secondary Control Enhancement Training; PAST = Parenting Adolescents Support Training; PL = Placebo; PT = Psychoanalytic Therapy; QIDS = Quick Inventory of Depressive Symptomatology; RADS = Reynolds Adolescent Depression Scale; RT = Relaxation Training; SER = Sertraline; (S)MFQ(-C) = (Short) Mood and Feelings Questionnaire (-Child); SSRI = Selective Serotonin Reuptake Inhibitor; TAU = Treatment As Usual; WL = Wait-list Condition.

**Table 5 ijerph-18-04600-t005:** Evidence base for adolescent depression treatment.

Review	Level 1 Well-Established	Level 2 Probably Efficacious	Level 3 Possibly Efficacious	Level 4 Experimental	Level 5 Questionable Efficacy
1998		Group CBT		Individual IPT FT	
2008	Group CBT Individual IPT	Individual CBT		Group IPT FT	
2016	Individual CBT Group CBT Individual IPT	Group IPT	Bibliotherapy CBT FT	Technology-assisted CBT	
Current	Individual CBT Individual IPT	Group CBT	FT	Bibliotherapy CBT Technology-assisted CBT Group IPT Short-term PT	

CBT = Cognitive Behavioral Therapy; FT = Family Therapy; IPT = Interpersonal Therapy; PT = Psychoanalytic Therapy.

## Data Availability

Data sharing is not applicable to this article.

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
