# Peer review of "Psychological Treatments for Depression in Adolescents: More Than Three Decades Later"

_ijerph, 2021, doi:10.3390/ijerph18094600_

Round 1

Reviewer 1 Report

Please, see attached file.

Author Response

We appreciate the excellent review and we consider the two questions raised very correct, which contribute to notably improve the article in two relevant aspects.

As the reviewer very well points out, according to the criteria adopted by Division 53 of the American Psychological Association, Society of Clinical Child and Adolescent Psychology, the paradox may arise that the ratio between positive / null results (even negative), of two psychological treatments differs substantially, despite which both could obtain the same status. This objection is particularly relevant with the individual TST, which reaches the status of probably effective treatment, despite the fact that there is only a single trial in which it has not been found to be superior to an alternative treatment. Following the reviewer's suggestion, we have added a new paragraph (line 692) to state the provisional nature of the probably effective treatment status of individual PT.

Regarding the second suggestion, we have relocated the paragraph on control conditions and made reference to the high rates of response to placebo and spontaneous recovery from depression in adolescents as an explanation for the fact that in some trials no significant differences were found with control conditions.

Reviewer 2 Report

First, I would like to congratulate the authors for their exhaustive systematic search of the literature and the detailed analysis of each of the studies reviewed. The manuscript provides a relevant contribution in addressing a research field that needs to be updated and a systematic literature review is a valid option. The review includes interesting information that can be of great value for the scientific community, especially for child and adolescent clinical psychology.

However, I suggest some clarifications and further elaborations before it fully meets the standards required for publication.

  1. TITLE, ABSTRACT AND INTRODUCTION.

  • The length of the abstract must be reduced to meet the journal’s requirements (maximum 200 words).
  • It might be interesting to specify the type of review performed in the keywords
  • The theoretical background in the introduction section is based on solid research and is adequately referenced, only minor revisions are suggested:
    • In lines 28-30, the authors mention comorbidity of depression with other psychological problems, is there a reference to support this statement?
    • In line 52, there is a spelling error ("ages" should be "aged").
    • Statements about clinical and epidemiological studies in line 58 should be supported with references.
    • The statement in line 95: "Despite its seriousness, depression often is underdiagnosed during adolescence, and there is great variability in the types of therapeutic approaches used." It is rather vague and not supported by scientific evidence, I would suggest rephrasing or providing adequate sources.
    • Although the objectives are clear and coherent with the introduction, it would be necessary to formulate the research question that warrants this review.

  1. METHOD.

  • Although the authors have used primary sources of access to studies that are relevant and valid for the purpose of the review, why didn’t they include the database of Web of Science?
  • Furthermore, information needs to be provided about the methodology followed in the review. Was this a systematic review that followed PRISMA standards?
  • What was the overall search process like? This information is especially interesting in the case of literature reviews (e.g. which key words did they use?).
  • How many researchers collaborated in the review process? If there were several, were they blinded? If so, I suggest calculating the level of agreement between raters or reviewers (e.g. Cohen’s Kappa) to increase the quality of the review.
  • If the review did not follow the quality standards for systematic reviews, it would be necessary to specify that this is a narrative literature review or rather a qualitative systematic review.

  1. RESULTS.
  • Following the recommendations of IJERPH: All Figures, Schemes and Tables should be inserted into the main text close to their first citation and must be numbered following their number of appearance (Figure 1, Scheme I, Figure 2, Scheme II, Table 1, etc.).
  • There are tables that are not mentioned in the text.
  • Tables should be numbered in order of appearance for example table 2 and table 4.
  • Table 5 should be included in the results section.
  • Although it is very interesting to report the reasons for exclusion of articles, perhaps it could be included as an annex (Table 2). There is an overwhelming excess of information and tables in the results section; it would be preferable to reduce or synthesize this section.

  1. DISCUSSION.
  • Why did the authors choose a review period from 1980 to 2020? The last review has been carried out in 2016, how does this review differ from previous ones and what new contributions have been made?
  • In the discussion, it would be convenient to reorganize the ideas and connect the different types of treatments. At times, the information seems redundant and repetitive with the results section.
  • Authors are requested to include limitations of the present review, not only of the studies included but also of the review itself.
  • In the conclusions section, please provide practical implications of this review.

  1. OTHER ASPECTS
  • Check the references section according to the journal guidelines (for example, the year should be written in bold).
  • Check that all in-text citations meet the required format (e.g. line 56).
  • Check also page numbers, some of them are incorrect.

Again, I congratulate the authors for their tremendous effort and extensive review.

Author Response

We appreciate the detailed and valuable review carried out, whose contributions improve the quality of the article. Following the reviewer's instructions:

  1. TITLE, ABSTRACT AND INTRODUCTION

- We have verified that the abstract does not exceed 200 words.

- We have specified the type of review in the keywords.

- The comorbidity of depression with other psychological problems is discussed in a specific paragraph (lines 57-65], where two bibliographic references are included.

- The statement about the underdiagnosis and the variety of therapeutic approaches to depression in adolescents is supported by the bibliographic reference [16]. To avoid possible confusion we have replaced the period and followed by a semicolon.

  1. METHOD

- We have corrected the omission and included the Web of Science database.

- We share the reviewer's considerations, but it is a qualitative review, carried out from a historical perspective, and not a systematic review, as is now required in the keywords (line 24) and indicated in the objectives of the review ( lines 101-104).

  1. RESULTS

- We have included in the text the mention of the tables in order of appearance.

- We agree with the reviewer that Table 2 contains a lot of information, but it seems convenient that the reader has it in the section of the review where it has been inserted.

  1. DISCUSSION

- The review has been chosen from 1980 because in that year research on depression in adolescents gained momentum when the American Psychiatric Association declared that “the essential features of a major depressive episode are similar in infants, children, adolescents and adults ”.

On the other hand, previous reviews, including the one from 2016, indistinctly reviewed treatment studies with adolescents diagnosed with a depressive disorder and indicated prevention studies with adolescents with depressive symptoms but without a diagnosis of depressive disorder. We have focused exclusively on the trials of psychological treatment of depression in adolescents, so we decided to review this extended period.

- We have reorganized the review headings so that there are now no separate sections for results and discussion.

- We have included a short final paragraph on the implications for professional practice and the limitations of the review.

  1. OTHER ASPECTS

- We have reviewed the formal aspects (references, numbering, etc.) of the text.

Reviewer 3 Report

Depression especially in Adolescents is a common and impairing disorder that is a serious public health problem in the world. The purpose of the current paper was to review the clinical trials conducted between 1980 and 2020 in adolescents with a primary diagnosis of a depressive disorder. This topic is interesting and important. And it was well written.

Comments:

1.This is a review. Why has the paper been devided into parts: “results” and “discussion". I can not find an obviously difference between “results” and “discussion”.

2.What is the role of Table 5. And did it belong to "results" part or "discussion" part?

3.Few research was found in Asian populations in this review. I want to know if there is few study in Asian populations or there is a low morbidity rate of pubertal depression in Asian populations?

Author Response

We appreciate the review made and agree with the comments on the structure of the article. As it is a qualitative review, with the aim of examining the efficacy of psychological treatments for depression in adolescents from a historical perspective, we agree with the reviewer that it does not conform to the structure of the research articles, so we have restructured the sections of the review.

With regard to research on psychological treatments for depression in adolescents in Asian populations, the review includes two trials with Chinese, Indian and Iranian populations, three trials conducted in the USA and the United Kingdom that included Asian minorities, in addition to four others unspecified ethnic minority trials. We believe that the rate of depression in natural teenagers in Asia need not be particularly low.

See for example:

Liu, X. C., Ma, D. D., Kurita, H. and Tang, M. Q. (1999). Self-reported depressive symptoms among Chinese adolescents. Social Psychiatry and Psychiatric Epidemiology, 34, 44-47.

Sun, X. J., Niu, G. F., You, Z. Q., Zhou, Z. K., and Tang, Y. (2017). Gender, negative life events and coping on different stages of depression severity: A cross-sectional study among Chinese university students. Journal of Affective disorders, 209, 177-181. (Age of participants: 16-25 years).